# Targeted Mutagenesis of the Multicopy *Myrosinase* Gene Family in Allotetraploid *Brassica juncea* Reduces Pungency in Fresh Leaves across Environments

**DOI:** 10.3390/plants11192494

**Published:** 2022-09-23

**Authors:** Dale Karlson, Julius P. Mojica, Thomas J. Poorten, Shai J. Lawit, Sathya Jali, Raj Deepika Chauhan, Gina M. Pham, Pradeep Marri, Sharon L. Guffy, Justin M. Fear, Cherie A. Ochsenfeld, Tracey A. (Lincoln) Chapman, Bruno Casamali, Jorge P. Venegas, Hae Jin Kim, Ashley Call, William L. Sublett, Lolita G. Mathew, Aabid Shariff, Joseph M. Watts, Mike Mann, Aaron Hummel, Ryan Rapp

**Affiliations:** Pairwise, 807 E Main St., Durham, NC 27701, USA

**Keywords:** CRISPR-CAS, *Brassica juncea*, leafy green, Cas12a, myrosinase, mustard bomb, glucosinolate, nutrition, targeted mutagenesis, biotechnology

## Abstract

Recent breeding efforts in *Brassica* have focused on the development of new oilseed feedstock crop for biofuels (e.g., ethanol, biodiesel, bio-jet fuel), bio-industrial uses (e.g., bio-plastics, lubricants), specialty fatty acids (e.g., erucic acid), and producing low glucosinolates levels for oilseed and feed meal production for animal consumption. We identified a novel opportunity to enhance the availability of nutritious, fresh leafy greens for human consumption. Here, we demonstrated the efficacy of disarming the ‘mustard bomb’ reaction in reducing pungency upon the mastication of fresh tissue—a major source of unpleasant flavor and/or odor in leafy *Brassica*. Using gene-specific mutagenesis via CRISPR-Cas12a, we created knockouts of all functional copies of the type-I myrosinase multigene family in tetraploid *Brassica juncea*. Our greenhouse and field trials demonstrate, via sensory and biochemical analyses, a stable reduction in pungency in edited plants across multiple environments. Collectively, these efforts provide a compelling path toward boosting the human consumption of nutrient-dense, fresh, leafy green vegetables.

## 1. Introduction

Vegetables are critical to food and nutritional security [1]. Green leafy vegetables (e.g., spinach, kale, mustard greens) are an exceptional source of vitamins, minerals, and phenolic compounds (https://fdc.nal.usda.gov/fdc-app.html#/food-details/1989450/nutrients, accessed on 19 April 2022). Mineral nutrients (e.g., calcium and iron) are more abundant in leafy vegetables than staple food grains [2], which comprise a greater portion of caloric intake (https://www.fao.org/3/u8480e/u8480e07.htm, accessed on 19 April 2022). The Centers for Disease Control and Prevention (CDC) recently reported that just 1 in 10 adults meet the daily intake recommendations for fruits and vegetables [3]. Multiple potential barriers exist for the consumption of fresh leafy green options, such as: cost, convenience, availability, and palatability/taste of the fresh produce. Among fresh leafy greens, lettuce is commonly used as a major component for salads and provides additive options for other dishes (e.g., additions on sandwiches, garnishes, etc.). Although lettuce is broadly available and commonly purchased by consumers, it lacks nutritional value relative to other leafy green vegetables [4].

In recent years, leafy green vegetables such as kale have gained popularity as ‘superfoods’ and serve as a nutrient-dense vegetable source for health-conscious consumers. In the Southern United States, turnips, mustard greens, and collards are a common part of the diet [5]. However, to minimize undesirable attributes associated with leafy greens, such as fibrous/tough leaves, bitterness, frilly textures, and/or pungency, consumers often cook down the leafy greens to soften tissues and alter flavor/odor profiles with the incorporation of fats and other ingredients. Consequently, the cooking process reduces the nutrient quality (e.g., folate and lutein) [6,7] and deprives consumers of the benefits of consuming fresh produce.

Among the most nutrient-dense leafy green options available [8], mustard greens (*B. juncea*) are characterized by intraspecific diversity with a variation in leaf traits, such as color, size, texture, and heading morphology [9,10,11]. If eaten fresh, mustard greens are pungent due to a reaction of the myrosinase enzyme with its glucosinolate substrates (Figure 1A) [12] in an effect triggered by chewing known as the ‘mustard bomb’ [13]. Consequently, mustard greens are either consumed fresh in smaller quantities as baby greens or are cooked to minimize pungency. Therefore, reducing the aforementioned ‘mustard bomb’ reaction and producing leafy greens with a reduced pungency in flavor and/or odor would enable a broader consumption of fresh, highly nutritious leafy greens. The advent of precision breeding technologies [14] such as CRISPR nucleases [15], base editors [16,17], and prime editing [18,19] has enabled opportunities for the rapid development of diverse and nutritious foods.

Here, we demonstrate the efficacy of disarming the ’mustard bomb’ reaction in reducing pungency upon mastication—a major source of unpleasant taste in leafy *Brassica* [20]. Using CRISPR-Cas12a gene editing techniques [21], we created loss-of-function alleles at all expressed multicopy type-I *myrosinase* genes in tetraploid *B. juncea* cv. Red Giant (Figure 1B,C). Greenhouse and field trials across multiple environments show that these plants are non-pungent via biochemical and sensory analyses. This demonstrates a compelling path toward improving human nutrition through the increased consumption of nutrient-dense fresh leafy green vegetables.

## 2. Results

### 2.1. Bioinformatics Analysis

The genome assembly of our *B. juncea* cv. Red Giant sequences produced a N50 of 1,549,726 bp, total size of 980,235,059 bp, and BUSCO [22] score of 99.4% complete BUSCOs with 89.7% duplicated, consistent with allo-tetraploid genome composition. From the *myrosinase* gene bioinformatics analyses, a total of 17 functional and expressed *B. juncea* myrosinases were identified as the most closely related enzymes to *Arabidopsis* type-I myrosinases (Figure 2). These genes are distributed in both the A and B genomes across seven chromosomes (Figure 3A). Predicted active site residues (Figure 3B) were identified and crRNA spacers were directed to disrupt the active site by targeting exon VI. The variation between *myrosinase* gene copies was circumvented by expressing seven crRNAs so that each copy was targeted by at least one crRNA spacer (Figure 3C).

### 2.2. Edit Characterization and Advancement of Edited Lines

After read mapping, the average coverage per gene was 1587 ± 719 (mean ± standard deviation). All 17 functional and expressed *myrosinase* genes were edited (Table 1). Of these, seven genes contained small deletions (≤25 bp), five contained large deletions (101 to 3781 bp), and the remainder were part of a large structural variation (Figure 4). The structural variant resided in a cluster of six *myrosinase* genes on chromosome B04 (only five of which were found to be expressed), where three *myrosinase* genes were deleted as part of 37 kb deletion and segments of three *myrosinase* genes were rearranged by combination of the 37 kb deletion and 100 kb inversion.

Plants were advanced at each generation based on the complement of inherited edits plus results of the glucose release assay (GRA; a biochemical measurement of myrosinase activity adapted from Dosz et al. [24]) and sensory data. A single first generation edited (E_0_) plant with desired edits and the lowest GRA value was advanced to the second (E_1_) generation. From the segregating E_1_ population, all plants determined to be transgene-free by PCR analysis, and that were homozygous for desired edits, were subsequently evaluated for pungency using a GRA assay. These plants were also evaluated by performing sensory analysis at baby (Figure 1A) and mature (Figure 1B) growth stages. Six individual non-pungent plants with an absorbance at 630 nm <0.5 were advanced to the third (E_2_) generation for further evaluation. Samples at the E_2_ generation were determined to be negative for plasmid DNA by the next generation sequencing of a library pulled down by a custom panel of probes homologous to the transformation plasmid sequence, a technique hereafter described as target-capture sequencing. The progeny of a plant with a high copy number of insertions of transformation plasmid sequence (>4) still known to contain transgenic elements was used as a positive control. The positive control, a 1:5 dilution and a 1:10 dilution of this sample, had a 31.0%, 33.3%, and 10.2% mapping rate to the plasmid, respectively, whereas non-transgenic, unedited samples (negative controls) had a 0.04% and 0.05% mapping rate to the plasmid (Appendix A). Plants without detectable transgene sequences were selected for further work. Thus, the state-of-the-art target-capture sequencing methodology used to probe genomic DNA for any component of the transformation plasmid revealed that the *B. juncea* lines with a reduced pungency do not have any foreign DNA integrated in its genome. Positive and negative controls provide confidence that this technique is equivalent in sensitivity and robustness to the methodology used in the characterization of heavily regulated transgenic product development [25,26].

The selected E_3_ line exhibited a significantly lower GRA than the unedited control, suggesting impaired myrosinase activity (Figure 5). Sensory analysis complemented the GRA data for the edited line, which was consistently non-pungent (score of 1 for edited vs. 3.7 for unedited; Appendix A) in taste evaluations across both growth stages in all tested environments. The E_4_ progeny from the line of interest was evaluated for edit and trait stability across multiple environmental conditions. Via the GRA, the baby and mature samples from the edited line were consistently differentiated from unedited Red Giant.

With our edited and control materials grown in multiple environments, baby and mature materials were harvested at different times within each location. Therefore, the growing conditions were different for the harvest stages within each environment, and it was expected that the GRA absorbance values could differ based on the growing environment for each harvest stage (Appendix A). This prediction was validated in the analysis based on a significant interaction between the environment and the harvest stage (*p*-value < 0.001) (Appendix A). The line effect was determined to be independent from the harvest stage and location. Therefore, the estimated difference between the unedited and edited lines is equivalent for both within the location and the growth stage (Appendix A).

### 2.3. Structural Modeling of Edited B05 Myrosinase

To determine the structural impact of the 21 bp deletion in the B05 *myrosinase* gene, both the unedited and edited proteins were modeled using the RosettaCM homology modeling protocol [27]. Rosetta’s Relax protocol was then used to allow each structure to reach an energy minimum independent of their template structures [28]. This deletion removes a seven-amino-acid sequence near the core of the protein (H200-Q206), shown in magenta in Figure 6A.

The deleted region contains a short beta strand and one position (Q206) that was previously noted to be important for catalysis [23]. In the unedited protein structure, this deleted segment is immediately followed by an alpha helix that also borders the proposed sinigrin binding site. In the edited protein, this alpha helix no longer forms, although a beta strand forms in the original deleted strand location (Figure 6B). To ensure that the strand’s formation was not a modeling artifact, its presence was independently confirmed by performing secondary structure prediction on the edited sequence using PSIPRED [29].

The original beta strand deletion and failure of the original helix to form places an arginine (R213) in the position originally occupied by Q206 (Figure 6C,D). This newly positioned arginine is predicted to interact with the active site residue Y351 and a nearby aspartate (Figure 6D).

MOLE2.5 [30] was used to further analyze the deletion’s impact on the binding pocket and identified potential tunnels with positions corresponding to previously determined catalytic residue locations [23]. Identified binding pockets are shown in Figure 6E (unedited) and Figure 6F (edited). When the binding pockets radii were measured (Appendix A), the unedited pocket was found to have a radius of 2 to 3 angstroms across the majority of its length. The edited myrosinase had a longer and narrower binding pocket, with the pocket radius being predominantly less than 2 angstroms.

## 3. Discussion

Significant phenotypic diversity exists within *B. juncea*, such as a variation in leaf shapes, colors, textures and phenological traits across cultivars. These visual and morphological traits make many *B. juncea* appealing to consider for fresh consumption. However, leafy *B. juncea* cultivars are commonly pungent when fresh leaves are consumed. This often strong pungency is a barrier for the broad adoption of fresh mustard greens by consumers. With the aim of making nutritious leafy greens more appealing for fresh consumption, we disabled the ’mustard bomb’ reaction within the selected cultivar. We used genome editing to produce non-transgenic and non-pungent *B. juncea*, which will be a source of high-quality nutrition for consumers.

The ‘mustard bomb’ reaction, evolved within some Brassicaceae species, is a specialized chemical reaction that is mediated by a spatially separated enzyme (myrosinase) and substrate pool (glucosinolates) [31]. Upon tissue damage (i.e., herbivory), myrosinases contact the glucosinolates (e.g., sinigrin) (Figure 1A) and various degradation products can be produced, some of which are strongly pungent (i.e., allyl isothiocyanate; AITC). Thus, this reaction has been proposed to have evolved as an anti-herbivory mechanism. Using a gene editing approach, we disarmed this reaction by knocking out myrosinase enzymes. After segregating away all transgenic elements, we demonstrated plants that were non-pungent when grown to baby and mature stages in a variety of locations and conditions. Opportunistic observations of herbivory across seasons and environments have not revealed an appreciable increase in insect feeding.

With limited functional knowledge of myrosinases in leafy mustard greens, we based our hypothesis and editing approach on functional studies that revealed that two type-I myrosinases (TGG1 and TGG2) were required to produce AITC in Arabidopsis [32]. To ensure that we targeted mutations with the greatest likelihood for the disruption of enzymatic properties, we performed sequence analysis relative to the known active site obtained from previously disclosed [33] three-dimensional structural analyses. This provided spatial context and enabled us to ensure that protein functionality would be disrupted by targeting Cas12a within a nucleotide sequence encoding a highly conserved and known functional enzymatic region.

Our editing approach recreates naturally occurring loss of function mutations in *myrosinase* genes. For example, our genomic analysis in *B. juncea* cv. Red Giant revealed a single gene copy with a mutation leading to a premature stop codon. We found that a *myrosinase* gene located at A09_cl02.3 has a TAG codon in exon 6 leading to a predicted premature stop. This results in a loss of the six succeeding exons, which include three of the six predicted active site residues in myrosinase as identified by Kumar and colleagues [23]. We found a similar mutation in another *B. juncea* cultivar, suggesting that this naturally occurring loss of function mutation is widespread. Efforts to recreate desired naturally occurring variants have been deployed in other crops, such as soybean, by creating lines with a high oleic trait. Using TALENs, Haun and colleagues [34] introduced deletions in FAD2 genes, resulting in an increase in oleic acid content from 20% to 80%. Such deletions are comparable to the naturally occurring variants in high oleic acid soybean lines carrying insertion/deletion mutations in FAD2 [35].

While most edited *myrosinase* genes in the described line contained large deletions or frameshifts (Table 1), the relatively small (21 base pairs) in-frame deletion’s impact on the B05 gene was less obvious. Since this deletion is near the protein core, and deletes a conserved secondary structure element, we originally hypothesized that it may have large structural impacts on the protein. However, the protein’s core and the proposed sinigrin binding site (Figure 6C) structure remain fairly conserved according to our predictive models. Therefore, we more closely examined the deletion’s effects on both potential substrate interactions and on the overall binding pocket shape. In the original binding pocket, the deleted glutamine is predicted, along with the catalytic glutamate (E423), to bind to the sinigrin molecule ring, while the negatively charged sulfate portion of sinigrin is predicted to interact with a nearby arginine, R280. Another arginine, R213, also neighbors the binding pocket near the expected sulfate position, possibly further stabilizing this negative charge.

In the edited myrosinase (Figure 6D), in addition to losing the substrate’s interaction with Q206, R213 is also displaced to be in Q206’s former position, closer to E423 and the predicted position of sinigrin’s ring rather than near the negatively charged sulfate. It appears to project farther into the binding site, and it is predicted to interact with nearby residues, including Y351, another sinigrin-interacting residue, possibly further restricting the binding pocket.

While both structures show a binding pocket in roughly the same position, the pocket identified in the edited myrosinase (Figure 6F) is visibly narrower in the center than the unedited protein pocket (Figure 6E), particularly where R213 protrudes into the pocket. This pocket narrowing, along with the loss of favorable interactions with Q206 and R213, predicts that this edited myrosinase is incapable of efficiently binding and processing sinigrin.

## 4. Conclusions

Advances in CRISPR technology have paved the way for the expansion of the use of desired causal variants in agricultural crops. Herein, we displayed the near total elimination of type-I myrosinase activity (Figure 5) via the targeted mutation of 17 gene copies of myrosinase in *B. juncea*, resulting in the elimination of pungency from fresh leaves. In as little as two generations, we also segregated away any foreign DNA used to eliminate the pungency from this nutritious leafy green, thereby positioning it to become a substantial source of vegetable servings for consumers. Relative to conventional breeding, the increased efficiency in delivering valuable traits will hasten the availability of phenotypic variation beneficial to farmers and consumers.

## 5. Materials and Methods

### 5.1. Genome Assembly of B. juncea cultivar Red Giant

A contig assembly was constructed for *B. juncea* cv. Red Giant using PacBio Hifi sequencing. DNA was extracted from dark-treated leaf samples with the DNAeasy Plant Pro Kit (Cat. No. 69206, Qiagen, Hilden, Germany). The standard SMRTbell library construction protocol (SMRTbell Express Template Prep Kit 2.0, Pacific Biosciences, Menlo Park, CA, USA) was performed with additional Damage Repair with NEB PrePCR Repair Mix (NEB, Ipswich, MA, USA) and a size selection step with Circulomics SRE (Circulomics, Baltimore, MD, USA) to remove short DNA fragments. Sequencing was performed on a Pacific Biosciences Sequel II instrument at the University of Georgia Genomics and Bioinformatics Core. From one SMRTcell, we generated 3,181,990 polymerase reads with a mean read length of 96,817 bp and a mean subread read length of 9904 bp. CCS software (https://github.com/PacificBiosciences/ccs, accessed 2 October 2020, version 4.2.0, Pacific Biosciences, Menlo Park, CA, USA) was used to generate 1,515,691 circular consensus sequence (CCS) reads with a mean read length of 11,581 bp. The CCS reads were then assembled using hifiasm version 0.12-r304 [36].

### 5.2. RNAseq Analysis

RNA was extracted from freshly harvested Red Giant leaves. RNA samples were quantified using Qubit 2.0 Fluorometer (Life Technologies, Carlsbad, CA, USA), and RNA integrity was assessed using an Agilent TapeStation 4200 (Agilent Technologies, Palo Alto, CA, USA). RNA sequencing libraries were prepared using the NEBNext Ultra II RNA Library Prep Kit for Illumina according to the manufacturer’s instructions (NEB, Ipswich, MA, USA). Briefly, mRNAs were first enriched with Oligo(dT) beads. Enriched mRNAs were fragmented for 15 min at 94 °C. First-strand and second-strand cDNAs were subsequently synthesized. cDNA fragments were end repaired and adenylated at 3’ends, and universal adapters were ligated to cDNA fragments, followed by index addition and library enrichment by limited-cycle PCR. The sequencing libraries were validated on the Agilent TapeStation (Agilent Technologies, Palo Alto, CA, USA), and quantified by using a Qubit 2.0 Fluorometer (Invitrogen, Carlsbad, CA, USA), as well as by quantitative PCR (KAPA Biosystems, Wilmington, DE, USA). The library was sequenced in paired-end mode, generating 77,807,572 reads. RNAseq reads were mapped with hisat2 [37], read counts per transcript were counted with htseq-count [38], and TPM values (transcripts per kilobase million) were calculated with a custom script to normalize and compare gene expression among samples.

### 5.3. Myrosinase Gene Bioinformatics

To identify putative functional and expressed myrosinase coding genes in the Red Giant assembly, a sequence search with BLAST [39] was performed using Arabidopsis *myrosinase* gene paralogs TGG1 (AT5G26000) and TGG2 (AT5G25980). BLAST hits were curated with a custom script to (1) filter out hits E value > 0.01 and query coverage < 70%, (2) merge nearby hits within 1000 bp to remove alignment gaps due to elevated sequence divergence within introns, and (3) retain sequences with at least 50% query coverage. Gene models from the automated genome annotation were manually curated using RNAseq sequence alignments. Gene copies were identified as likely pseudogenes when the coding sequence mutations led to stop codons or coding sequence exons were missing. Expression analyses were applied to identify gene copies that were expressed in Red Giant leaves. Gene copies with TPM > 0.5 were characterized as expressed. A multiple sequence alignment was performed with Clustal Omega v.1.2.2 [40] and a neighbor joining phylogenetic tree was constructed with Geneious Prime (https://www.geneious.com, accessed on 19 April 2022, version 2020.0.4, Dotmatics, Boston, MA, USA) using HKY distance model and 500 bootstrap replicates to estimate node support.

### 5.4. Vector Design

The binary plasmid, pWISE687, was designed to express the *Lachnospiraceae* bacterium ND2006 Cas12a (LbCas12a) endonuclease [21] and 7 guide RNA spacers (Pwg120225: GTGGAAAGGTAAAGCACTGGATC, Pwg120226: GTGGAAAGGTGAAGAACTGGATC, Pwg120227: GTGACAAAGTAAAGCACTGGTTC, Pwg12028: GTCCTGTAAAGATCGACGACCGT, Pwg120229: CGCCGTAACACCTGTGCTTGGTA, Pwg120230: CGCCGTAACATCTCTCATCAACC, Pwg120231: GAGAACATCGACCAGGTGCATCT) to knock out the *myrosinase* genes in *B. juncea* (Figure 3B,C). Spacers were designed to mutate Exon VI for the following reasons: (1) it is the first exon with more than one active site, thus increasing the probability of creating loss of function for a specific ortholog; (2) this exon exists in the first 1-3 exons of the *Brassica nigra* and *Brassica rapa* myrosinase sequences—suggestive of conserved importance for this region for enzyme function; (3) targeting Exon VI allows us to deploy the least number of spacers to target the greatest number of myrosinase gene copies. Multiple sequence alignments of the exon were manually curated in the target regions for each spacer sequence and the manually curated alignments were used to generate the sequence logos in Geneious to illustrate variation in the target regions across all myrosinase loci. Guides 1, 2, and 3 were driven by the GmU6i promoter [41] and guides 4, 5, 6, and 7 were driven by the AtU6-26 promoter [42]. Transcription of guide RNAs was terminated with the polyT terminator (TTTTTTT). The guide cassettes were synthesized by GenScript and inserted in the binary plasmid with the LbCas12a expression cassette. A disarmed *Agrobacterium tumefaciens* strain was used to introduce a T-DNA from the binary plasmid. The T-DNA cassette expresses a plastid-targeted *aadA* expression cassette as a plant selectable marker, the cassette for LbCas12a endonuclease, and 2 guide RNA cassettes. The binary plasmid was re-extracted from *Agrobacterium* and verified by whole plasmid sequencing by plexWell PRO™ (seqWell, Beverly, MA, USA).

### 5.5. Plant Transformation

*B.**juncea* cv. Red Giant seeds (Kitazawa Seed Co., Salt Lake City, UT, USA) were surface sterilized with 70% ethanol for 30 s and washed three times with sterile reverse osmosis (RO) water followed by treatment with 20% bleach containing Tween 20 for 20 min. The seeds were then washed three times with sterile RO water, transferred to sterile filter paper, and blotted, dried, and arrayed on plates containing Murashige and Skoog’s basal media [43] for germination. The cultures were incubated at 25 °C under a 16 h photoperiod for 4 to 8 days. The seedling-derived explants were excised and inoculated with a suspension of *Agrobacterium*, carrying pWISE687, at an OD600 of 0.5, followed by sonication for 30 s. The sonicated explants were kept in *Agrobacterium* suspension for 30 min, followed by *Agrobacterium* suspension removal and blotting dry the explants. Approximately 25–30 explants were plated on inoculation and co-culture media and the cultures were incubated at 20–23 °C for 3 days under a 16 h photoperiod. The explants were moved to shoot induction media. The cultures were refreshed every 2–4 weeks on the same media until shoot induction was observed from the transformed tissues. The shoots were subsequently transferred to shoot elongation and rooting media to induce roots. The rooted plants were then acclimatized in a controlled environment prior to the transfer of plugs.

### 5.6. Controlled Environments and Field Growth Conditions

Initial plantlets (E_0_ generation) were plugged onto FlexiPlug^®^ 72-R Slice plugs (Quick Plugs, Falmouth, MA, USA), on a 36-cell flat. Flats were placed in a growth chamber for 35 days and covered with a clear plastic dome for the initial seven days. Growth chamber environmental conditions were 16 h day length, 22/18 °C day/night temperature, 50% humidity, and 350 µmol·m^−2^·s^−1^ light intensity. Plantlets were watered daily through subirrigation. At day 35 after plugging, plantlets were transplanted into 17.8 cm diameter pots (2.78 L) filled with BM7 substrate (Berger, Saint-Modeste, QC, Canada) and fertigated until drainage with deionized water + 200 ppm N of 15-5-15 Cal Mg N-P-K fertilizer. Plants were then grown under a 16 h day length, 23/18 °C day/night temperature, 50% humidity, and 350 µmol·m^−2^·s^−1^ light intensity. Upon initial signs of bolting, plastic pollination bags were used to cover entire plants to avoid cross-pollination and to ensure self-pollination. Plants were grown within pollination bags until seed harvest.

E_1_ seeds derived from the E_0_ plants were sown onto 36-cell flats filled with PGX substrate (Pro-Mix^®^, Quakertown, PA, USA). Plants were grown under similar conditions to E_0_ plants, modified only with an increase in light intensity to 500 µmol·m^−2^·s^−1^ after transplanting. Seeds were harvested after ~120 days after sowing. The same growth conditions in controlled environment were used from E_0_–E_3_ generations.

Non-transgenic E_3_ seeds (4th generation) were primed with CaCl_2_ [100 mM] and incubated for 18 h in the dark with constant shaking @110 rpm. After a 48 h drying period, seeds were placed on filter paper imbibed with 500 ppm GA_3_ solution inside petri dishes. Seeds were maintained in dark conditions for 72 h at 20 °C to promote germination. On 29 April 2021, plants were transplanted to 60-cell trays filled with germination mix substrate. Flats were then covered with clear plastic domes and 70% shade cloth and placed inside a hoop house with cooling pads. Five and seven days after transplanting, shade cloth and plastic domes were removed, respectively. Plants were grown in the hoop house until 26 May 2021, being watered daily with well-water amended with 100 ppm N 15-5-15 Cal Mg N-P-K. On 26 May 2021, transgene-free plants were transplanted to the field inside isolation cages to ensure purity of seed production in Moses Lake, WA, USA. Myrosinase edited *Brassica juncea* was deemed to not be regulated via the USDA Am I Regulated process [44]. In the field, plant spacing was 12 cm in-row by 60 cm between rows. Plants were drip-irrigated daily during establishment and irrigation was reduced to as-needed during the growing cycle. Plants were bulk-harvested on 26 August 2021 in each cage separately. Location soil type is Ephrata gravelly sandy loam. Monthly maximum, average, and minimum temperatures and humidity, and precipitation are shown in Appendix A. Fertility and pesticide applications were performed according to the recommended guidelines for mustard green production in the area.

E_4_ seeds (5th generation) were sown directly in the open field on 15 November 2021, in Yuma, AZ, USA. Two different areas per line were sown, one at a planting density of 7.9 million seeds per hectare to harvest baby leaves and another at a planting density of 247,000 seeds per hectare to harvest mature leaves. Irrigation was applied immediately upon sowing to ensure proper seed imbibition. Additional irrigation events occurred approximately once a week during the growing period. Baby leaves were harvested on 4 January 2022 and mature leaves were harvested on 2 February 2022. Location soil type is Kofa clay soil type. Monthly maximum, average, and minimum temperatures and humidity, and precipitation are shown in Appendix A. Fertility and pesticide applications were carried out following the recommended guidelines for mustard green production in the area.

### 5.7. Deep Sequencing of Edited Plants

Editing efficiencies and sequence profiles were characterized by deep sequencing regions of the *myrosinase* genes targeted in this study. Genomic DNA was isolated from leaf tissues using the PureGenome Plant isolation kit (Aline biosciences, Woburn, MA, USA) according to the manufacturer’s instructions. Genomic DNA quantity and quality was measured using a NanoDrop 8000 spectrophotometer (Thermo Fisher Scientific, Waltham, MA, USA). Briefly, 10 ng of DNA was used as template for PCR. Multiplex gene specific forward and reverse PCR primers were designed (Appendix A) to span *myrosinase* gene cut sites, generating around 471–510 bp products in unedited plants. NGS Amplicon libraries were generated using a two-step PCR method, where primary PCR with 5′ tails allows a secondary PCR to add Illumina i5 and i7 adapter sequences and barcodes for multiplexing of several hundred samples. PCR amplifications were performed using the following parameters: 98 °C for 30 s; 28 cycles for PCR1 and 8 cycles for PCR2 (98 °C 10 s, 55 °C 20 s, 72 °C 30 s); 72 °C for 5 min; hold at 12 °C. The PCR reactions were performed with Phusion Green High-Fidelity DNA Polymerase (Thermo Fisher Scientific, Waltham, MA, USA). The secondary PCR amplicon samples were individually purified using AMPure XP beads according to manufacturer’s instruction (Beckman Coulter, Brea, CA, USA). All purified samples were quantified using a Spectra Max ID3 plate reader (Molecular Devices, San Jose, CA, USA), pooled with an equal molar ratio and run on an AATI fragment analyzer (Agilent Technologies, Palo Alto, CA, United States). The pooled amplicon libraries were sequenced on an Illumina MiSeq (2 × 250 paired end) using a MiSeq Reagent kit v2 (Illumina, San Diego, CA, USA).

### 5.8. Probe Design for Myrosinase Gene Characterization and Transgene Presence/Absence Using Target-Capture Sequencing

Custom probe panels for gene characterization and to assess the presence or absence of transgenic plasmid elements using target-capture sequencing were designed and synthesized by Twist (Twist Biosciences, San Francisco, CA, USA). The gene target capture (GTC) panel was developed against the 18 type-I *myrosinase* genes (~143 probes), and the plasmid panel was designed against a library of transformation plasmids (~100 probes). Two positive control genes for ANAC038 and ARF4 were included in the GTC and plasmid panel design (ANAC038: BjuB003410, ARF4: BjuO010408) [45]. Probe panels were optimized for efficient enrichment of targets using 120 bp double stranded oligonucleotide probes that provide uniform coverage of the regions’ double stranded DNA probes and facilitate capture of both strands of genomic DNA. The tilling coverage for GTC was at 0.25× and plasmid was at 1×. Plants were determined to be transgene-negative so long as the percent mapping rate to the plasmid sequence was well below a background noise threshold of 1% (Appendix A). The positive control loci (ANAC038: BjuB003410, ARF4: BjuO010408) [45] showed similar mapping rates across samples, indicating consistent enrichment for targeted DNA fragments (Appendix A).

### 5.9. Target-Capture Library Preparation and Sequencing

Library preparation and the target-capture method were performed using the standard Twist protocol. Genomic DNA was isolated from leaves using Mag-Bind Plant DNA 96 kit (Omega Bio-Tek, Norcross, GA, USA). The DNA was quantified by using a Qubit 4 fluorometer (Thermo Fisher Scientific, Waltham, MA, USA). Indexed libraries were prepared by using Twist Library Preparation Enzymatic Fragmentation (EF) Kit 2.0 (Twist Biosciences, San Francisco, CA, USA). Approximately 100 to 300 ng was fragmented, end repaired, dA-tailed, and ligated with universal adapters, followed by cleanup of fragmented DNA with AMPure XP beads (1× ratio) (Twist Biosciences, San Francisco, CA, USA). Adapter-ligated fragments were amplified for 11 cycles with Twist UDI primers (Twist Biosciences, San Francisco, CA, USA) and the size distribution of library fragments was measured on the fragment analyzer. Eight individual libraries were combined into one pool by adding equal concentration of individual library (not exceeding more than 4 μg of total DNA). Capture pools were hybridized with the custom probe panel for 16 h at 70 °C and then incubated with streptavidin beads following the standard Twist workflow (Twist Biosciences, San Francisco, CA, USA). The post-capture library was amplified for 12 cycles and pair-end sequenced on the Illumina MiSeq platform (Illumina, San Diego, CA, USA), generating 2 × 250 bp paired-end reads.

### 5.10. Bioinformatics Procedure of Plasmid Target-Capture Sequencing for Plasmid Purity Test

NGS reads were pre-processed by (1) clipping the first and last 25 bases of each read-pair, (2) trimming adapter sequences and low-quality bases, and (3) removing reads < 35 bases long. We mapped the reads to a combined reference that included *B. juncea* cv. Red Giant genome assembly and the plasmid sequence using BWA-mem [46]. The number of mapped reads were counted for the plasmid sequence and the two positive control genes in the *B. juncea* genome. The positive control genes were assessed to ensure that the target capture procedure worked properly to enrich DNA fragments with homology to the capture probes. Samples were identified as negative for plasmid DNA when the percent mapping rate to the plasmid sequence was below a background noise threshold of 1%. The 1% threshold was determined by comparing the percentage of reads mapping to the plasmid sequence in unedited samples and in test samples where nuclease presence was assessed with qPCR vs. transgenic samples (Appendix A). Plasmid-negative samples had reads mapping between 0 and 0.5%, whereas plasmid-positive samples showed read mapping rates between approximately 10 and 60%.

### 5.11. Bioinformatics Procedure of Gene Target-Capture Sequencing for Edit Characterization

NGS reads were pre-processed by (1) clipping the first and last 25 bases of each read-pair, (2) trimming adapter sequences and low-quality bases, and (3) removing reads < 35 bases long. We mapped the reads to the *B. juncea* cv. Red Giant genome assembly using BWA-mem [46]. Sequencing data quality was assessed by the total number of reads mapping to the *B. juncea* genome and average coverage on gene targets. We set a minimum average per-gene coverage of 100×. Small variants were called with Freebayes v1.3.4 [47] and structural variants were called with LUMPY [48]. Per-sample genotypes and allele depths were extracted into a table format. Minimum per-variant depth was set to 30× to confidently call homozygous genotypes. Read support of edit calls was confirmed by visual inspection of read alignments in IGV [49].

### 5.12. Phenotypic Evaluation

Homozygous-edited, transgene-free plants were grown in different environmental conditions to characterize the stability of the non-pungent trait. The edited lines were grown in greenhouse conditions within Pairwise (Durham, NC, USA) facilities and under field conditions in Salinas (CA, USA) and Yuma (AZ, USA) following standard agronomic practices for leafy greens and as described above.

Pungency was measured on leaf samples across all three growing conditions at both baby and mature stages using two approaches: (a) GRA that indirectly measures the activity of myrosinase enzyme; and (b) leaf sample sensory evaluation for pungency, using a trained internal panel. For all evaluations, leaf samples were randomly selected from harvested material. Baby leaves were defined as having an average leaf length of 10 cm and width of 5 cm, and mature leaves were defined as having an average leaf length of 25 cm and average leaf width of 20 cm. In addition to pungency, the edited lines were also evaluated for morphological attributes, such as leaf size, shape, and plant architecture, to ensure similarity relative to the unedited control.

### 5.13. GRA Assay

Myrosinase activity was measured using a glucose release assay (GRA) adapted from Dosz et al. [24]. Briefly, three 4 mm leaf punches were sampled for three to six individual leaves from each sample. The leaf punches from each sample were collected in a single tube of a 96-well matrix plate using the 4 mm Integra™ Miltex™ Biopsy Punches with Plunger System (Fisher: 12-460-410, Integra™, Mansfield, MA, USA). To each tube within the collection microtubes (19560, Qiagen, Hilden, Germany), 500 μL of 10 mM (-)-sinigrin hydrate (85440-1G, Sigma-Aldrich, St. Louis, MO, USA) and two Lysing matrix S. beads (Fisher: 6925000, MP Biomedicals™, Solon, OH, USA) were added and the plant material was macerated using a Geno/Grinder (SPEX SamplePrep 2010, Metuchen, NJ, USA) (1750, 30 s). A tube with the buffers and no leaf material was used as a negative control; unedited plant samples were incorporated as positive biological controls. The macerated material was incubated in a water bath at 40 °C for 30 min, flash frozen in liquid nitrogen, and stored at −80 °C. To assay samples, they were thawed in a water bath set at 40 °C for 2 min and immediately centrifuged for five min at maximum RPM in swing-out rotor centrifuge. Supernatant aliquots of 30 μL were transferred from each sample into a single well of a 96-well plate. Standards were 30 μL glucose solutions (continuous 1:2 dilutions starting with 50 mM D-glucose) into respective wells. To these wells, 200 μL of ABTS-glucose solution [2.7 mM ABTS, 1000 units of peroxidase (Type VI-A, P6782 Sigma-Aldrich, St. Louis, MO, USA) and 1000 units of glucose oxidase in 100 mL] were added using a multichannel pipette. The 96-well plates were then incubated for 15 min at room temperature. Immediately following this period of incubation, the absorbance of samples at 630 nM was measured using a PowerWave HT microplate spectrophotometer (Agilent Bio-Tek, Winooski, VT, USA). To account for any potential residual glucose in leaf extracts outside of myrosinase activity, parallel reactions were run in 500 μL of water without any buffers, enabling the opportunity for comparative analyses to specifically identify the responsiveness of myrosinases to exogenous sinigrin substrate.

### 5.14. Sensory Assay for Evaluating Pungency

A sensory panel of 4–7 members evaluated the materials for pungency. The panelists refrained from eating/drinking (other than water) 30 min prior to attending the session. For tasting each sample, a bitesize section of the leaf sample (approximately 5 × 5 cm) was selected, rinsed in water, and gently blotted dried on a paper towel. The leaf tissue was masticated and pungency in mouth was assessed on a 5-point scale and designated as non-pungent (score of 1), slightly pungent (score of 2), moderately pungent (score of 3), pungent (score of 4), and very pungent (score of 5). Palettes were rinsed with water and crackers between each sample, and next sample was evaluated after all flavor/sensations dissipated from the oral cavity.

### 5.15. Structural Modeling of Unedited and Edited Myrosinase Encoded on Chromosome B05

Homology models of the unedited and edited myrosinase encoded on chromosome B05 were produced using the RosettaCM homology modeling protocol [27]. Homologues were first identified using the HH-suite [50,51] as follows: for each sequence, the HHblits program was run for two iterations on the Uniprot90 database to create a hidden Markov model (HMM). Secondary structure predictions were added to each HMM, and resulting models were used to identify homologues from a database of sequences with known structures filtered to 70% maximum identity using the HHsearch program. A custom script was used to select a limited number of templates with high similarity to the input structure.

These homologues, along with Rosetta fragment files produced using the recommended protocol [52], were then used to produce homology models for each sequence using RosettaCM. For the unedited myrosinase sequence, a representative low-energy model was selected from the first five models produced. For the edited myrosinase sequence, 100 models were produced and divided into no more than 10 clusters, giving priority to the most energetically stable structures [53]. The most stable of these clusters was selected for further analysis. For each input sequence, the selected structure was then used as an input for Rosetta’s FastRelax protocol [28]. Ten structures were produced for each input model, and the lowest-energy structure was used for further analysis.

Secondary structure predictions were performed on both the unedited and edited sequences of B05 myrosinase using PSIPRED [29] using the same parameters used for Rosetta’s fragment-picking algorithm [52]. To characterize the binding pockets in each model, MOLE2.5 [30] was used to identify tunnels within the lowest-energy edited and unedited models using a 0.5 angstrom bottleneck radius, with all other settings at default values. Visualizations were performed using the PyMOL Molecular Graphics System (The PyMOL Molecular Graphics System, Version 2.5.2 Schrodinger, LLC New York, NY, USA)

### 5.16. Statistical Analysis

A statistical comparison was conducted for the difference in the edited and unedited lines. GRA data were analyzed with an analysis of variance model (ANOVA) with factors for the cultivar, harvest stage, environment, and associated interactions. A backward model selection method was employed to establish a final model. Effects were removed from the model if the *p*-value was greater than 0.05. The significance of main effects was evaluated if the effect was not present in an interaction. Interactions for cultivar by harvest stage and cultivar by environment were removed from the model since they were not found to be significant predictors of absorbance values. The ANOVA model assumption of normality was reviewed both graphically and using Shapiro–Wilk [54] and Kolmogorov–Smirnov [55] normality tests. There were no notable outliers or normality concerns evident. Cultivar comparisons were conducted within each harvest stage and environment utilizing Tukey multiple testing correction. The statistical analysis and associated graphs were generated using the R programing language (version 4.0.3) [56,57,58,59,60,61].

## Figures and Tables

**Figure 1 plants-11-02494-f001:**
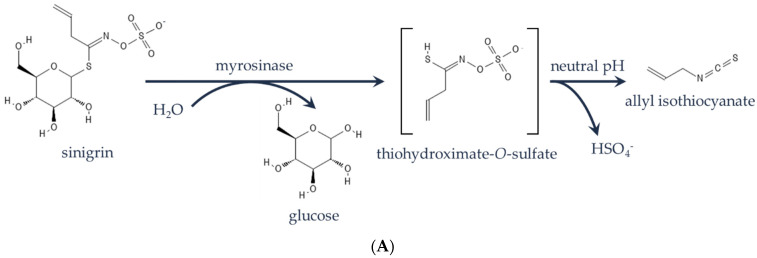
The ‘mustard bomb’ reaction (**A**) with sinigrin as the glucosinolate hydrolyzed by myrosinase. *Brassica juncea* cv. Red Giant (**B**) baby (average leaf length of 10 cm and width of 5 cm) and (**C**) mature plants (average leaf length of 25 cm and average leaf width of 20 cm).

**Figure 2 plants-11-02494-f002:**
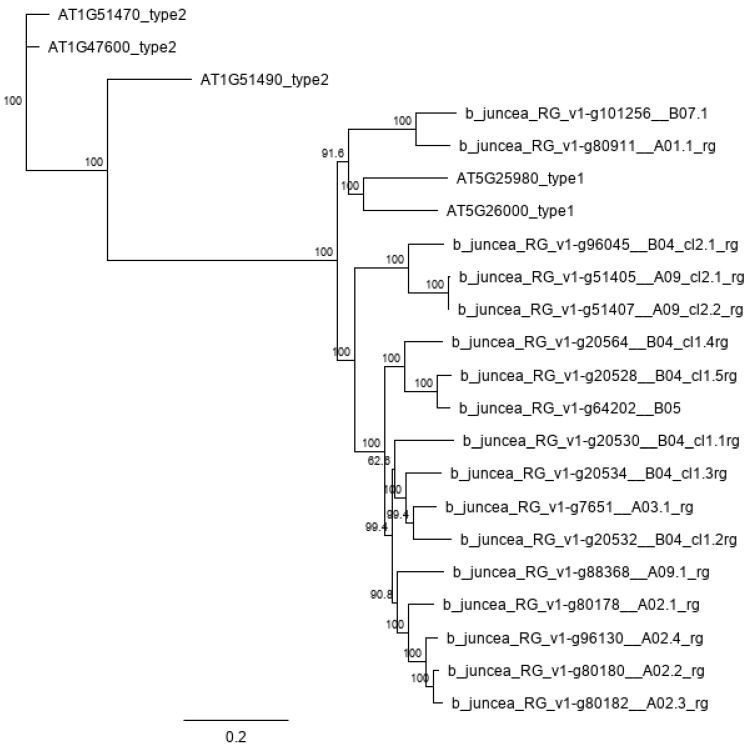
Phylogeny of 17 type-I *myrosinase* genes in *Brassica juncea*.

**Figure 3 plants-11-02494-f003:**
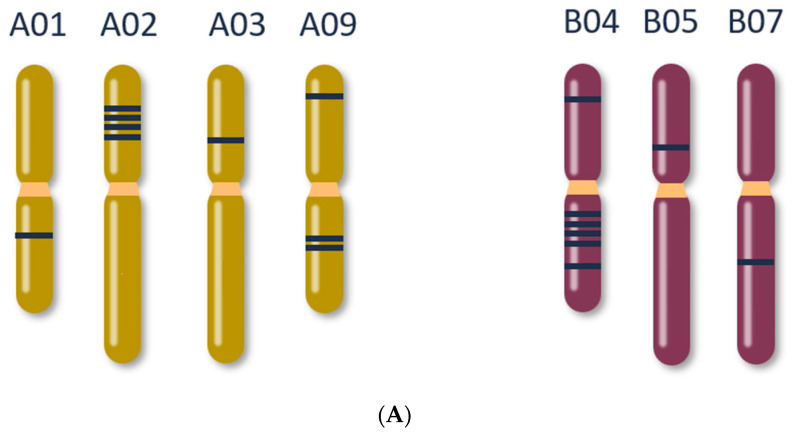
Genomic location of 17 type-I *myrosinase* genes in *Brassica juncea*. (**A**) Distribution of *myrosinase* genes across the A and B genomes. (**B**) Representative gene model for *myrosinase*. Yellow bars within green exons represent the six predicted active site residues in myrosinase as identified by Kumar and colleagues [23]. Editing spacers pair to sequences in Exon VI (regions 1, 2, 3, 4). (**C**) Sequence logos representing regions where guide sequences were designed in Exon VI of myrosinase. Alignments were manually curated. (1) PWg120225, PWg120226, PWg120227; (2) Pwg120231; (3) PWg120229, PWg120230; (4) PWg120228.

**Figure 4 plants-11-02494-f004:**
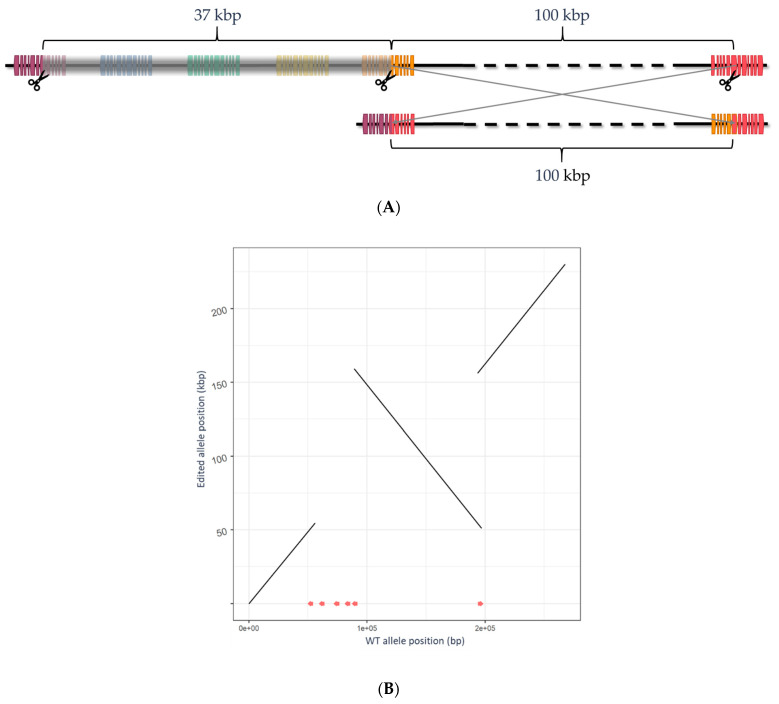
The structural variation produced in the cluster of six *myrosinase* genes on chromosome B04. (**A**) Schematic diagram of the structural variation produced (intergenic regions not to scale). The top segment shows the original genomic structure and three evident Cas12a cut sites (scissors). The bottom segment shows the structural variation comprising a 37 kbp deletion spanning three entire genes and a 100 kbp inversion resulting in two chimeric genes both with premature stops in exon 6. B04_cl1.1_rg (green); B04_cl1.2_rg (yellow); B04_cl1.3_rg (orange); B04_cl1.4_rg (red); B04_cl1.5_rg (purple); B04_cl1.6_rg—not expressed (blue); 37 kbp region deleted low-lighted in gray. (**B**) Dot plot showing ~270 kb region of chromosome B04 with structural variant in edit allele (*y*-axis), including ~100 kb inversion and ~37 kb deletion. Myrosinase gene models are shown with red line segments along the *x*-axis.

**Figure 5 plants-11-02494-f005:**
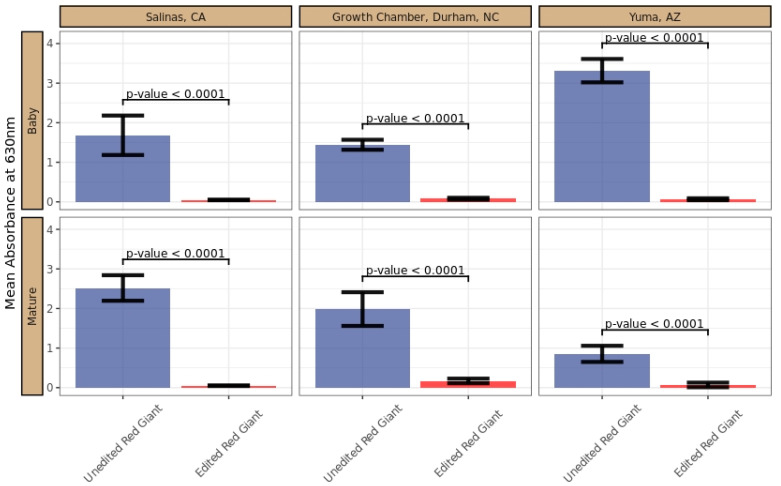
Glucose release assay of unedited vs. edited *Brassica juncea*. Mean glucose release absorbance (GRA) values at 630 nm from leaf discs in the presence of sinigrin for unedited and edited Red Giant mustard cultivars by environment and harvest stage (Appendix A). Error bars represent the standard error for the mean GRA value. The *p*-value is for the post hoc two-tailed pairwise comparison between the edited and unedited cultivar (Appendix A). Leaf samples were randomly selected from harvested material. Data values are the GRA colorimetric absorbance values at 630 nm from leaf discs processed in the presence of sinigrin.

**Figure 6 plants-11-02494-f006:**
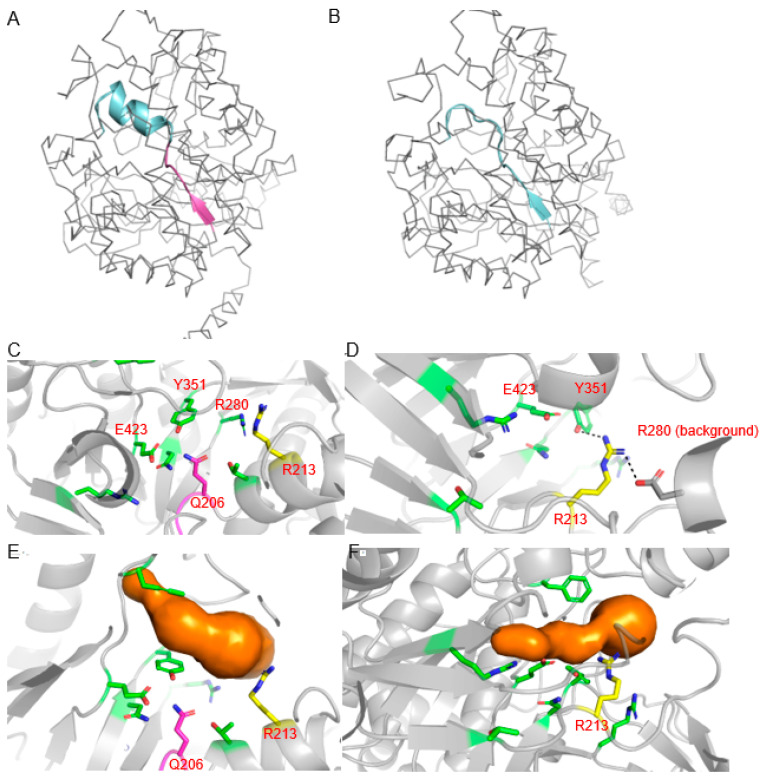
Overview of structural effects of the 21 bp deletion on myrosinase. In native myrosinase (**A**), the deleted region corresponds to a short beta strand and loop (magenta) that is immediately followed by a small alpha helix (cyan). Upon original beta strand and loop deletion, the former alpha helix (cyan) instead forms a new beta strand in the original strand’s position in place of an alpha helix (**B**). This deletion removes one of several catalytic residues, Q206 (magenta), previously identified in docking studies [23], and displaces R213 (yellow) formerly located on the downstream alpha helix (**C**). In the edited structure (**D**), this arginine is shifted closer to the catalytic glutamate and is predicted to extend further into the binding pocket; it is also predicted to interact with the sinigrin-binding residue Y351. When potential binding tunnels were identified using MOLE2.5 [30], the identified binding pocket (**E**) appears to narrow in the center in the edited protein (**F**) near the new position of R213.

**Table 1 plants-11-02494-t001:** Edit allele characterization at myrosinase loci in the homozygous E_n_ Red Giant line.

Gene	Edit	Putative Edit Effect
A01.1_rg	7 bp deletion	Pre-mature stop (at exon 6)
A02.1_rg	365 bp deletion	Loss of at least 2 active site residues; pre-mature stop (at exon 7 exons 5 and 6 partially deleted)
A02.2_rg	1923 bp deletion	Loss of multiple exons upstream of the spacer (exons 1–5 deleted, exon 6 partially deleted)
A02.3_rg	10 bp deletion, 3971 bp deletion	Loss of multiple exons downstream of the spacer (exon 6 partially deleted, exons 7–12 deleted)
A02.4_rg	93 bp deletion, 8 bp deletion	Pre-mature stop (at exon 6)
A03.1_rg	7 bp deletion, 7 bp deletion	Pre-mature stop (at exon 6)
A09.1_rg	8 bp deletion, 13 bp deletion, 4 bp deletion	Pre-mature stop (at exon 7)
A09_cl2.1_rg	8 bp deletion	Pre-mature stop (at exon 6)
A09_cl2.2_rg	8 bp deletion, 6 bp deletion	Pre-mature stop (at exon 6)
B04_cl1.1_rg	whole gene deletion	Whole gene deletion
B04_cl1.2_rg	whole gene deletion	Whole gene deletion
B04_cl1.3_rg	B04 gene chimera via inversion	Pre-mature stop (at exon 6)
B04_cl1.4_rg	B04 gene chimera via inversion	Pre-mature stop (at exon 6)
B04_cl1.5_rg	B04 gene chimera via inversion	Pre-mature stop (at exon 6)
B04_cl2.1_rg	316 bp deletion	Loss of 1 active site residue
B07.1_rg	8 bp deletion, 5 bp deletion	Pre-mature stop (at exon 7)
B05	21 bp deletion	7 aa deletion including one active site residue (Q>K in exon 6)

## Data Availability

The data presented in this study are available on request from the corresponding author. The data are not publicly available due to third-party obligations, their proprietary nature, and potential intellectual property.

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
