# Peer review of "Targeted Mutagenesis of the Multicopy Myrosinase Gene Family in Allotetraploid Brassica juncea Reduces Pungency in Fresh Leaves across Environments"

_plants, 2022, doi:10.3390/plants11192494_

Round 1

Author Response

  1. The discussion needs to be more elaborated. It doesn’t talk a lot about the recent literature reports. Specifically, how much percentage of myrosinase activity was reduced in edited cells when compared to non-edited and why?
    • The authors feel that paragraphs 4-7 of the discussion adequately describe why myrosinase activity was eliminated. Specifically in paragraph 5, we begin “While most edited myrosinase genes in the described line contained large deletions or frameshifts (Table 1), the relatively small (21 base pair) in-frame deletion’s impact on the B05 gene was less obvious.” We go on for several paragraphs explaining how the 21 bp in-frame mutation does eliminate function.
    • The conclusion of the discussion was elaborated, to read: Advances in CRISPR technology pave the way to expand the use of desired causal variants in agricultural crops. Herein, we have displayed the near total elimination of Type I myrosinase activity (Figure 5) via targeted mutation of 17 gene copies of myrosinase in juncea, resulting in elimination of pungency from fresh leaves. In as little as two generations, we also segregated away any foreign DNA used to eliminate the pungency from this nutritious leafy green, thereby positioning it to become a substantial source of vegetable servings for consumers. Relative to conventional breeding, the increased efficiency in delivering valuable traits will hasten the availability of phenotypic variation beneficial to farmers and consumers.

  1. The author seemed to have used a lot of different methods but the data representation seems a little hazy. For example: can they include a southern blot analysis to show the transgenic lines (to represent table 1 better)?
    • Greater clarity is needed to address the general comment of “data representation seems a little hazy.” Rebutting the example provided, Table 1 does not describe transgenic lines, but rather edits of 17 different gene copies in a single line (this fact was further clarified in the header for Table 1. We have not performed Southern Blot analysis to evaluate the transgene present in the E0 generation, but rather utilized target capture to analyze the genome comprehensively and digitally for any presence of heterologous fragments resulting from any portion of the entire plasmid used for T-DNA insertion from the Agrobacterium. As cited in the manuscript (Zastrow‐Hayes, G.M.; Lin, H.; Sigmund, A.L.; Hoffman, J.L.; Alarcon, C.M.; Hayes, K.R.; Richmond, T.A.; Jeddeloh, J.A.; May, G.D.; Beatty, M.K. Southern‐by‐Sequencing: A Robust Screening Approach for Molecular Characterization of Genetically Modified Crops. Plant Genome 2015, 8, eplantgenome2014.08.0037, doi:10.3835/plantgenome2014.08.0037.) this is the accepted state of the art.

Reviewer 2 Report

Authors might want to obtain low level of GSLs by editing Myrosinase gene and successfully obtained the mutants. Many researchers focused on reduction the bitter tasted or toxic components and increasing useful GSLs in Brassica crops. This work also tried to edit the multicopy genes, which is one of the most difficult jobs. So, I wish authors to be more careful in drawing conclusions.

1.     gRNAs were designed in only exon VI. Authors mentioned 6 active site in figure 3. Why did you chose only one region? It would be better to explain why authors made that choice.

2.     In line 111 of page 6, “All E1 plants determined to be transgene free by PCR analysis” Authors should show the data not mention. Also, authors should describe how E1 plants did not have plasmid vector.

3.     Authors presented the information of mutation in Table 1. However, mutation rate in genome and positions on chromosome should be presented. Among 17 numbers of Myrosinase gene, which member was changed or not changed? These information is important to determine if the enzyme lose the function or not/

4.     Section 5.14 of M&M, author described the sensory assay. However, I couldn`t find the data. Authors should show the data to describe reduction of pungency.

5.     In line 131 of page 6 and line 271 of page 13, authors mentioned “data not shown” and “data in preparation”. I think authors show the data or remove the sentences.

6.     Authors should describe the meanings of supplementary data, especially Table s5 and s6.

7.     In figure s5, authors mentioned the “baby leaves”. However, the leaves were harvested 31 days post wet date. What is the total growing duration? Why did the author call a 31-day-old plant a baby?

Author Response

  1. gRNAs were designed in only exon VI. Authors mentioned 6 active site in figure 3. Why did you chose only one region? It would be better to explain why authors made that choice.
    • The following rationale was added to the Materials and Methods: “Guides were designed in Exon VI because of multiple reasons: 1) it is first exon with more than one active site thus increasing the probability of creating loss of function for a specific ortholog; 2) this exon exists in the first 1-3 exons of the Brassica nigra and Brassica rapa myrosinase sequences– suggestive of conserved importance for this region for enzyme function; 3) targeting Exon VI allows us to deploy the least number of spacers to target the greatest number of myrosinase gene copies.”
  2. In line 111 of page 6, “All E1 plants determined to be transgene free by PCR analysis” Authors should show the data not mention. Also, authors should describe how E1 plants did not have plasmid
    • In the fourth line of the second paragraph of section 2.2 “Results: Edit characterization and advancement of edited lines” it was further clarified that not all E1 plants were free of plasmid, but that all which were free of plasmid (and were homozygous for desired edits) were further evaluated.
  3. Authors presented the information of mutation in Table However, mutation rate in genome and positions on chromosome should be presented. Among 17 numbers of Myrosinase gene, which member was changed or not changed? These information is important to determine if the enzyme lose the function or not.
    • The second and third columns of Table 1 show the mutations found in each of the 17 gene copies described all of which were changed. The third column describes the putative edit effect on gene function. Since there is no public genome for Brassica juncea Red Giant, we do not see the value in presenting the gene positions further than what is provided (the chromosome location). Similarly, the mutation rate for the individual genes would pertain to the E0 population created and further do not bear relevance to the selected E1 and greater generation materials evaluated.
  4. Section 5.14 of M&M, author described the sensory However, I couldn`t find the data. Authors should show the data to describe reduction of pungency.
    • The data is added to the manuscript in the results section under ‘Edit characterization and advancement of edited linessection
  5. In line 131 of page 6 and line 271 of page 13, authors mentioned “data not shown” and “data in preparation”. I think authors show the data or remove the sentences.
    • The sentences have been deleted.
  1. Authors should describe the meanings of supplementary data, especially Table s5 and .
    • Additional text was added to the headers for Tables s5 and s6. Figure s6 was removed for simplification.
  2. In figure s5, authors mentioned the “baby leaves”. However, the leaves were harvested 31 days post wet date. What is the total growing duration? Why did the author call a 31- day-old plant a baby?
    • The ‘baby leaves’ used in the manuscript is reflective of the size of the leaves (approx. 2in. Wide x 4in. Long) more than the age of the plant and aligns with the terminology used in salad industry.

Round 2

Reviewer 2 Report

  1. Section 5.14 of M&M, author described the sensory However, I couldn`t find the data. Authors should show the data to describe reduction of pungency.
    • The data is added to the manuscript in the results section under ‘Edit characterization and advancement of edited lines’ section

          à Do you have the data set? Authors wrote down the values in manuscript.

       I suggest to insert the data set(table or figure).

  1. In line 131 of page 6 and line 271 of page 13, authors mentioned “data not shown” and “data in preparation”. I think authors show the data or remove the sentences.

·          

    • The sentences have been deleted.

è  I found one more “data in preparation line” 137 of page 6.

Authors should check and change (or remove) the these mentions.

3.     In figure s5, authors mentioned the “baby leaves”. However, the leaves were harvested 31 days post wet date. What is the total growing duration? Why did the author call a 31- day-old plant a baby?

    • The ‘baby leaves’ used in the manuscript is reflective of the size of the leaves (approx. 2in. Wide x 4in. Long) more than the age of the plant and aligns with the terminology used in salad industry.

   à Authors should insert these descriptions in main manuscript (M&M) for readers who treat the different plant materials.

Author Response

Dear Ms. Bianca Wu and Reviewers,

My co-authors and I have revised the manuscript and provide the below responses. We sincerely hope that you will find these revisions adequate.

Kind regards,

Shai Lawit and co-authors

REVIEWER #1

  1. Section 5.14 of M&M, author described the sensory However, I couldn`t find the data. Authors should show the data to describe reduction of pungency.
    • The data is added to the manuscript in the results section under ‘Edit characterization and advancement of edited linessection

  • → Do you have the data set? Authors wrote down the values in manuscript.

  • Table s2 was added to summarize the sensory data.
  1. In line 131 of page 6 and line 271 of page 13, authors mentioned “data not shown” and

“data in preparation”. I think authors show the data or remove the sentences.

  • The sentences have been deleted.

  • ➔ I found one more “data in preparation line” 137 of page 6. Authors should check and change (or remove) the these mentions.

  • This sentence was deleted. The authors searched the document and found no other occurrences.

  1. In figure s5, authors mentioned the “baby leaves”. However, the leaves were harvested 31 days post wet date. What is the total growing duration? Why did the author call a 31- day-old plant a baby?
    • Response 1: The ‘baby leaves’ used in the manuscript is reflective of the size of the leaves (approx. 2in. Wide x 4in. Long) more than the age of the plant and aligns with the terminology used in salad industry.

  • ➔ Authors should insert these descriptions in main manuscript (M&M) for readers who treat the different plant materials.

  • We should have pointed out previously that the descriptions can be found in Materials and Methods: Phenotypic evaluation. The language has been slightly modified to further clarify.